# HEV ORF2 protein-antibody complex deposits are associated with glomerulonephritis in hepatitis E with reduced immune status

Anne-Laure Leblond [1,12], Birgit Helmchen[1,12], Maliki Ankavay[2], Daniela Lenggenhager [1], Jasna Jetzer [1], Fritjof Helmchen [3], Hueseyin Yurtsever[4], Rossella Parrotta[1], Marc E. Healy[1], Amiskwia Pöschel[5], Enni Markkanen [5], Nasser Semmo[6], Martin Ferrié[7], Laurence Cocquerel [7], Harald Seeger [8], Helmut Hopfer[9], Beat Müllhaupt[10], Jérôme Gouttenoire [2], Darius Moradpour[2], Ariana Gaspert[1] & Achim Weber [1,11] ✉

Hepatitis E virus (HEV) infection, one of the most common forms of hepatitis worldwide, is often associated with extrahepatic, particularly renal, manifestations. However, the underlying mechanisms are incompletely understood. Here, we report the development of a de novo immune complex-mediated glomerulonephritis (GN) in a kidney transplant recipient with chronic hepatitis E. Applying immunostaining, electron microscopy, and mass spectrometry after laser-capture microdissection, we show that GN develops in parallel with increasing glomerular deposition of a non-infectious, genome-free and non-glycosylated HEV open reading frame 2 (ORF2) capsid protein. No productive HEV infection of kidney cells is detected. Patients with acute hepatitis E display similar but less pronounced deposits. Our results establish a link between the production of HEV ORF2 protein and the development of hepatitis E-associated GN in the immunocompromised state. The formation of glomerular IgG-HEV ORF2 immune complexes discovered here provides a potential mechanistic explanation of how the hepatotropic HEV can cause variable renal manifestations. These findings directly provide a tool for etiology-based diagnosis of hepatitis E-associated GN as a distinct entity and suggest therapeutic implications.

Hepatitis E virus (HEV) infection, one of the most common causes of acute hepatitis, is a major global health problem[1-3]. The predominantly enterically transmitted HEV infection has two main epidemiologic patterns that correlate with geographically prevalent HEV genotypes[4]. In resource-limited countries, endemic and epidemic HEV-1 and −2 are transmitted mainly through contaminated drinking water. In resource-rich countries, zoonotic HEV-3 and −4 infections predominate,

transmitted mainly through contaminated meat products. Despite its high prevalence in industrialized countries, HEV-3 infection has been underdiagnosed in Europe and North America for many years, in part because of its highly variable clinical presentation[2,5,6]. The spectrum ranges from an asymptomatic course to acute, self-limiting hepatitis to acute-on-chronic liver failure in patients with pre-existing liver disease and chronic hepatitis in immunocompromised individuals[2,3,7].

---

HEV-3 infection in particular has been associated with extrahepatic mostly neurological and renal manifestations. While underlying pathomechanisms are still largely unknown[8], extrahepatic manifestations are expected to develop either directly, i.e., by HEV infection of the respective organs or indirectly, i.e., by immunologic reactions[8–11]. Therefore, it is conceivable that, apart from renal injury generally associated with impaired liver function, kidney dysfunction in hepatitis E may be caused—solely or additionally—by HEV-inherent mechanisms. Histologically confirmed glomerular diseases reported in patients with hepatitis E including membranoproliferative glomerulonephritis (MPGN), with or without cryoglobulinemia, and membranous GN, argue for an underlying immune-mediated mechanism[8,12–15]. However, a direct pathophysiologic link to HEV infection, proving a causal relationship with hepatitis E, has not yet been established[8].

Central to the understanding of HEV pathogenesis is the genetic organization and life cycle of this positive-strand RNA virus whose genome harbors three main open reading frames (ORF) encoding ORF1 non-structural proteins with viral replicase function, the ORF2 protein, corresponding to the capsid protein and main antigenic structure[9,16], and the ORF3 protein involved in viral particle secretion[17]. As described in human serum and in vitro cell models, HEV produces different ORF2 isoforms with distinct molecular weights: a non-glycosylated intracellular isoform (ORF2intra) assembled into infectious particles (ORF2i) and glycosylated isoforms (ORF2g/c) secreted in large amounts exhibiting 3 sites of glycosylation in positions 137, 310 and 562[18–20].

Here, we describe the development of de novo immune complex-mediated glomerulonephritis associated with the glomerular deposition of HEV ORF2 protein in a patient with chronic hepatitis E, and similar but less pronounced deposition in patients with acute hepatitis E.

## Results

### Gradual development of immune complex glomerulonephritis with membranoproliferative pattern in a kidney transplant recipient with hepatitis E

Detailed clinical information on Patient 1 is provided in (Fig. 1a) and in the Supplementary file. Autopsy findings in Patient 1 included liver cirrhosis, hepatitis with necrosis, and hepatocytes immunohistochemically positive for the HEV ORF2 protein, confirming hepatitis E. Kidney transplant histology showed persistent proliferative and sclerosing immune complex-mediated glomerulonephritis (GN) with a membranoproliferative pattern (MPGN), consistent with MPGN with immune complexes (IC-MPGN)(Fig. 1b), which had been diagnosed in a more subtle form in kidney transplant biopsies taken four and three months before death (Fig. 1c). There was no evidence of recurrent IgA nephropathy or antibody-mediated rejection. Remarkably, the renal allograft as well as retrospectively examined previous biopsies with GN showed strong immunohistochemical staining for HEV ORF2 protein, decorating the peripheral capillaries and the mesangium of all glomeruli (Fig. 1b, right, and 1c). This indicated virus replication for at least 4 months, thus establishing the diagnosis of chronic hepatitis E[2]. Re-evaluation of the patient's previous graft biopsies showed a progressive course: initial subtle mesangial expansion, mild hypercellularity, and immune complex deposition, progressing to a membranoproliferative pattern with endocapillary hypercellularity and significant subendothelial deposits. Reticular aggregates were found in the cytoplasm of endothelial cells at the last graft biopsy. Prolonged GN was associated with a markedly increased immunohistochemical (IHC) reactivity for the HEV ORF2 protein. This was paralleled by worsening renal function and increasing proteinuria (Fig. 1a). Subendothelial electron dense deposits were confirmed by electron microscopy, which also showed subepithelial and mesangial deposits but no particles suggestive of virions (Fig. 1c).

### No evidence of productive HEV infection of kidney cells

These observations prompted us to search for any signs of infection in the kidney cells themselves, as recently suggested[11]. Quantification of the viral load and visualization of HEV RNA by RNA in situ hybridization (ISH) confirmed the on-going infection in the liver (frozen and FFPE materials) (Fig. 2). For heart, brain and spleen, there was no evidence of viral replication as HEV RNA (data not shown for ISH) or HEV ORF2 protein positive cells were not visualized in these tissues (Supplementary Fig. 1 for IHC). The low viral load measured for these organs likely resulted from the high viremia of Patient 1. In contrast, there was no detection of HEV RNA in the kidney, either by ISH or qRT-PCR. Thus, although our negative finding did not exclude a direct infection of kidney cells, we had no evidence of viral replication in kidney tissue. Overall, these findings suggested that HEV ORF2 protein is a genome-free form, i.e., not associated with HEV virions.

### A 60 kDa HEV ORF2 protein formed immune complexes within glomeruli

The initial nephropathologic examination confirmed the diagnosis of de novo immune complex glomerulonephritis (IC-GN) with deposition of immune complexes within glomeruli, assessed by IgG and C3 positivity in basement membrane and mesangium. The deposition of extracellular HEV ORF2 protein shared the same spatial distribution within glomeruli, indicative of HEV-associated IC-GN (Supplementary Fig. 2a and Supplementary Table 1).

To corroborate this finding, we next performed co-localization studies and confirmed that extracellular colocalization of IgG with HEV ORF2 protein was statistically highly significant (p < 10E-10; Fig. 3a and Supplementary Fig. 2b). HEV ORF2 immune complexes were not detected in other organs examined (brain, spleen, heart) (Supplementary Fig. 1). Immunoblot analysis using monoclonal antibody 1E6 on liver and kidney tissue extracts revealed a band at around 60 kDa, with a stronger signal in the kidney than in the liver (Fig. 3b). To verify this finding of a truncated HEV ORF2 protein, we sequenced the protein contents of liver and kidney by mass spectrometry (MS). Mass spectrometry on liver protein extracts of Patient 1 confirmed the presence of HEV ORF2 protein (Supplementary Table 2). For kidney, we combined laser-capture microscopy (LCM) and MS to distinguish the protein content of the interstitial and glomerular compartments (Supplementary Table 2). LC-MS/MS analysis of glomeruli revealed, among other fragments, the glomerular marker podocin and HEV ORF2 protein, notably containing the epitope recognised by 1E6 antibody (Figs. 3c, d). In contrast, podocin was not detected in the interstitium, and HEV ORF2 protein only in traces (4 hits versus 241 in the glomerular compartment; Supplementary Table 2). This argued for 1) sufficient differential preparation of glomerular versus interstitial compartments, and 2) deposition of HEV ORF2 protein almost confined to the glomeruli.

Collectively, these findings demonstrated that the glomerular isoform corresponds to a truncated 60 kDa HEV ORF2 protein, mostly located within glomeruli, where it forms immune complexes.

### Characterization of the glomerular HEV ORF2 protein

Based on Western blot (WB) and IHC results, we showed that the truncated glomerular HEV ORF2 protein contains the epitope region of the 1E6 antibody (amino-acids, aa 437–457).

To identify the other regions preserved in the sequence of the glomerular HEV ORF2 protein, especially the biologically relevant glycosylation sites[21,22], we first performed WB with 1E6 antibody after glycosidase treatment using PNGase F. No change in molecular weight was observed in kidney protein extract with or without treatment. This result means that the kidney isoform is not glycosylated, because either the glycosylation site(s) are not glycosylated or truncated (Fig. 4a). We then took advantage of recently available monoclonal antibodies (mAbs) specific for non-glycosylated and infectious HEV

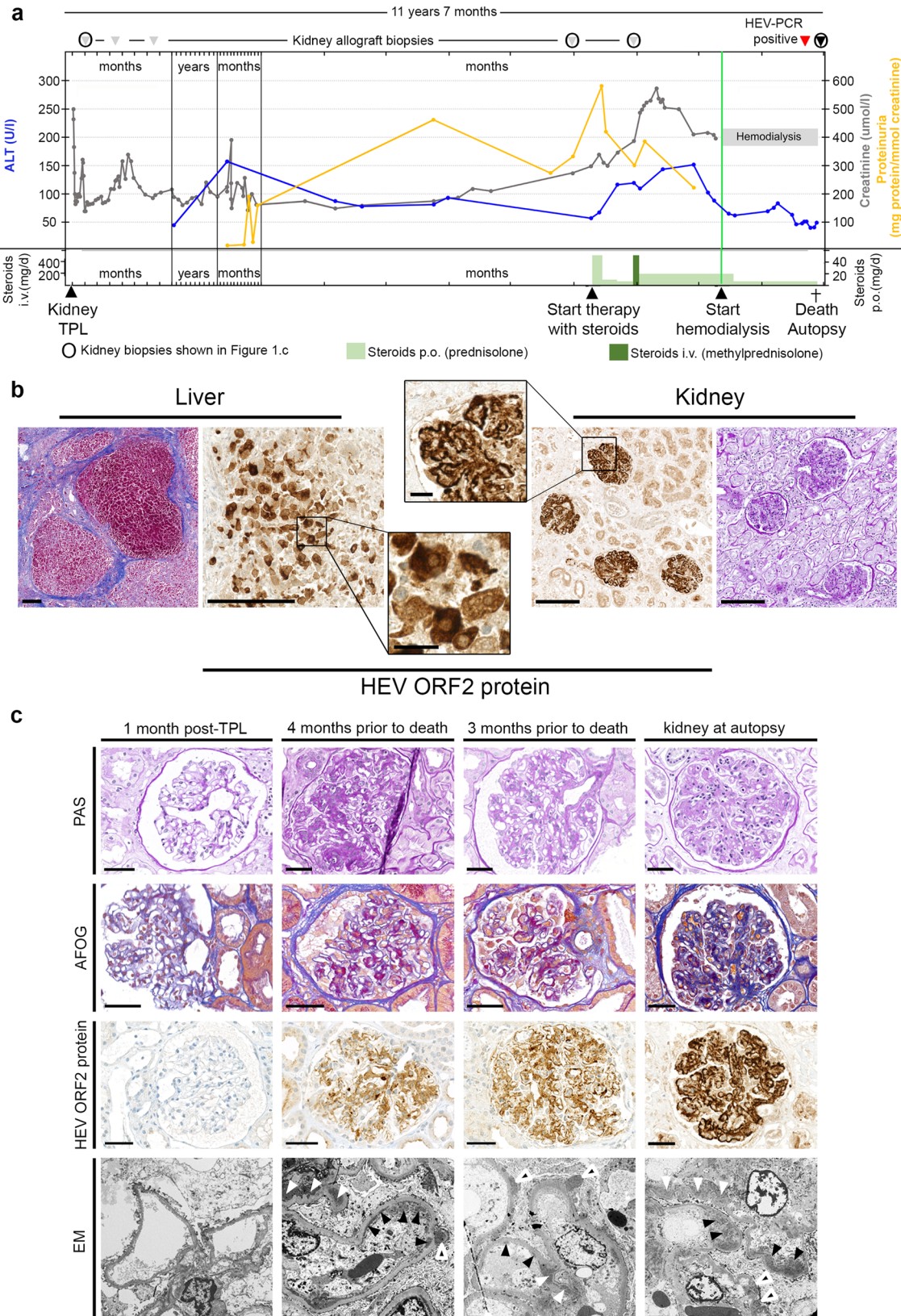

**b** Liver / Kidney

HEV ORF2 protein

**c**

ORF2i protein: P1H1 that is specific for the N-terminal of the non-glycosylated infectious HEV ORF2i protein (its epitope covers aa 23–36) and P2H1/H2 that target the glycosylation site in position 562 (N3) if it is not occupied (their epitopes cover aa 555–567)[22]. Unlike in HEV-replicating cells and the liver tissue, IHC with these mAbs led to negative results in the glomeruli, suggesting that the epitopes of these

mAbs are absent (Fig. 4b). Finally, after on-slide deglycosylation, IHC with P2H1 mAB remained also negative, confirming the absence of the N3 glycosylation site (Supplementary Fig. 4).

Collectively, these findings demonstrate that 1) the glomerular HEV ORF2 protein is not glycosylated, and 2) it lacks the N-terminal region of the infectious HEV ORF2i protein and the region of the N3 site

**Fig. 1 | Clinical course, autopsy findings and gradual development of immune complex glomerulonephritis with membranoproliferative pattern in a kidney transplant recipient with hepatitis E. a** Course of alanine transaminase (ALT, blue), proteinuria (yellow) and creatinine (grey), time points of therapeutic and diagnostic interventions as well as death / autopsy. **b** Histology of autopsy liver showing cirrhosis (Masson trichrome stain) and hepatitis with immune reactivity for HEV ORF2 protein in hepatocytes (left). Histology of transplant kidney at autopsy showing glomerulonephritis (periodic acid Schiff [PAS] stain) and (extra-cellular) immune reactivity for HEV ORF2 protein in glomeruli (right). Scale bars in overviews, 200 μm; scale bars in detail, 25 μm. Stainings performed more than 5 times on different tissues blocks. **c** Kidney histology. One month post transplantation: inconspicuous glomeruli on light microscopy (PAS and acid fuchsin-Orange G [AFOG] stains), no HEV ORF2 protein deposits, no electron dense deposits on electron microscopy (EM). Four months prior to death (biopsy 4): glomerulus with mild mesangial and endocapillary hypercellularity, segmental sclerosis and prominent podocytes (PAS stain). Mostly mesangial and few glomerular basement membrane protein deposits (AFOG stain). Moderate mesangial and glomerular basement membrane positivity for HEV ORF2 protein. Mesangial (white arrowheads), subendothelial (black arrowheads) and subepithelial (black and white arrowheads) on EM. Three months prior to death (biopsy 5): glomerulus with mild mesangial and endocapillary hypercellularity (PAS stain). Mostly mesangial and few glomerular basement membrane protein deposits (AFOG stain). Moderate to strong mesangial and glomerular basement membrane positivity for HEV ORF2 protein. Mesangial (white arrowheads), subendothelial (black arrowheads) and subepithelial (black and white arrowheads) on EM. Kidney at autopsy: glomerulus with mild mesangial and endocapillary hypercellularity (PAS stain). Mostly mesangial and few glomerular basement membrane protein deposits (AFOG stain). Strong mesangial and glomerular basement membrane positivity for HEV ORF2 protein. Mesangial (white arrowheads), subendothelial (black arrowheads) and subepithelial (black and white arrowheads) on EM. Scale bars in PAS, AFOG, and HEV ORF2 protein images: 50 μm; scale bars in EM images: 2 μm. Source data are provided as a Source Data file.

---

of glycosylation[21,22]. Based on these results, the glomerular isoform corresponds to the sequence region comprised between aa 36-554, leading to an estimated molecular weight of 57 kDa, corroborating our WB finding of about 60 kDa. It is therefore conceivable that the glomerular isoform still contains the N1 (position 137) and N2 (position 310) glycosylation sites, but that these are not glycosylated.

### Glomerular HEV ORF2 protein deposits in patients with acute hepatitis E

These findings prompted us to examine kidney tissue from additional hepatitis E patients. In three identified cases, all of whom had pre-existing liver cirrhosis and died of acute-on-chronic liver failure in a context of acute hepatitis E[23,24], we found deposits similar as in Patient 1, albeit at lower intensities (Table 1). Subtle proliferative glomerular changes with IgG/HEV immune complexes (vizualised by co-immuno-fluorescence) were detected, consistent with early hepatitis E-associated GN (Supplementary Fig. 3).

## Discussion

Here we describe and characterize glomerular immune complexes containing strong deposition of HEV ORF2 protein in the setting of chronic and, to a lesser extent, acute HEV infection. To determine the significance of the glomerular IgG/HEV ORF2 protein deposits in relation to renal dysfunction associated with hepatitis E[10,13], it is important to consider both host and HEV characteristics. Impaired renal function has been reported for HEV-1 and -3, and associated with both acute and chronic hepatitis E[7,10,25], generally more transient and milder in the acute form[8]. Its variable presentation includes 1) clinically silent urinary excretion of HEV ORF2 protein with preserved kidney function[26] 2) transiently impaired kidney function (with or without proteinuria) with resolution after normalization of transaminases[12,25], and 3) subclinical or overt de novo immune complex glomerulonephritis (GN) with variable outcome including kidney failure[8,12].

The development of GN seems to be associated with an impaired immune status[15]. Accordingly, Patient 1 who developed membrano-proliferative GN (MPGN) with bona fide IgG/HEV ORF2 protein deposits was immunocompromised. However, reduced immuno-competence can also be assumed for patients 2-4, who all had liver cirrhosis[27]. The well-documented temporal association between HEV infection and renal disease, together with the quantitative correlation between ORF2 protein levels and impairment of renal function, argue for a causal relationship between glomerular deposits and renal dysfunction[12,26].

Infection-related immunopathogenesis is known as a possible cause of GN. It occurs either via in situ immune complex formation as in poststreptococcal GN, via deposition of circulating immune complexes as e.g., in HCV, HIV and EBV or via direct cytotoxic effects of pathogens as for example in HIV, EBV and SARS-CoV-2[28]. Chronic viral infections, such as HCV and HBV, with or without circulating cryo-globulins, are an important cause of GN with membrano-proliferative pattern (MPGN)[15]. In 1993, HCV was recognized as a common cause of immune-complex-mediated MPGN (IC-MPGN)[29,30]. There is no animal model for a HCV-induced GN, in which the immune-complex glomer-ulonephritis is directly caused by an infection of the experimental animal with the HCV. However, there is a mouse model that closely mimics HCV-induced MPGN. In this model, a systemic inflammation occurs affecting, among other organs, the kidney. These mice con-sistently develop glomerular disease in form of MPGN that closely resembles that seen in humans[31,32].

We have observed the same type of immune complexes in both acute and chronic hepatitis E, but more pronounced in the latter, in accordance with significantly higher HEV ORF2 protein levels found in sera from chronically as compared to acutely HEV-infected individuals[33]. In patients 2-4, renal dysfunction was due to hepato-renal syndrome. Nevertheless, it is conceivable that in addition to cases of fully developed glomerulonephritis[8,12–14], as in Patient 1, glo-merular damage associated with more subtle HEV-ORF2 protein deposits, as in Patients 2-4, who can also be regarded as immunocompromised[27], may represent a general early morphological correlate and harbinger of impaired glomerular function in the context of HEV infection[25].

We suggest a causal relationship for HEV and IC-MPGN due to biological plausibility and coherence: in particular 1) the co-localization of HEV ORF2 protein with IgG and C3 in the glomerular lesions, accompanied by increased proteinuria, and 2) the finding of a biological gradient[34].

Based on our observations, we cannot deduce whether HEV ORF2 protein-associated immune-mediated glomerular damage also occurs in acute or subclinical hepatitis E in immunocompetent individuals, in the course of which (transiently) impaired renal function has also been described[4,10]. However, the host immune status determines the dura-tion of HEV persistence and thus indirectly also the amount of HEV ORF2 protein formed cumulatively[7,33]. Several factors potentially determine the development and the severity of GN, including genetic and immunologic on the host side. If the immune status is the decisive factor determining the extent of glomerular damage, it is expected to be lower in acute than in chronic hepatitis E, in line with our obser-vations. However, a larger cohort would be necessary to investigate the candidate factors that are likely to determine whether and to what extent hepatitis E-associated GN develops.

HEV antigens can remain detectable in sera of patients with chronic hepatitis E > 100 days after clearance of HEV RNA, emphasizing that the presence of HEV ORF2 protein does not necessarily correlate with infectious virions[33]. It has been shown that HEV exists in urine not

## a

| Organ | Specimen | Material | HEV genome copy number/ug total RNA | HEV ORF2 protein detection |
|--------|----------|----------|--------------------------------------|----------------------------|
| **Liver** | Autopsy | FF | $4.4 \times 10^6$ | ++ |
| | | FFPE | $9.1 \times 10^6$ | |
| **Kidney** | Autopsy | FF | not detected | ++ |
| | | FFPE | not detected | |
| **Heart** | Autopsy | FF | $1.3 \times 10^4$ | - |
| | | FFPE | not detected | |
| **Brain** | Autopsy | FF | $1.2 \times 10^4$ | - |
| | | FFPE | not detected | |
| **Spleen** | Autopsy | FF | $2.3 \times 10^4$ | - |
| | | FFPE | not detected | |

FFPE: Formalin-Fixed Paraffin-Embedded, FF: Fresh Frozen
not detected: Ct value >35

## b

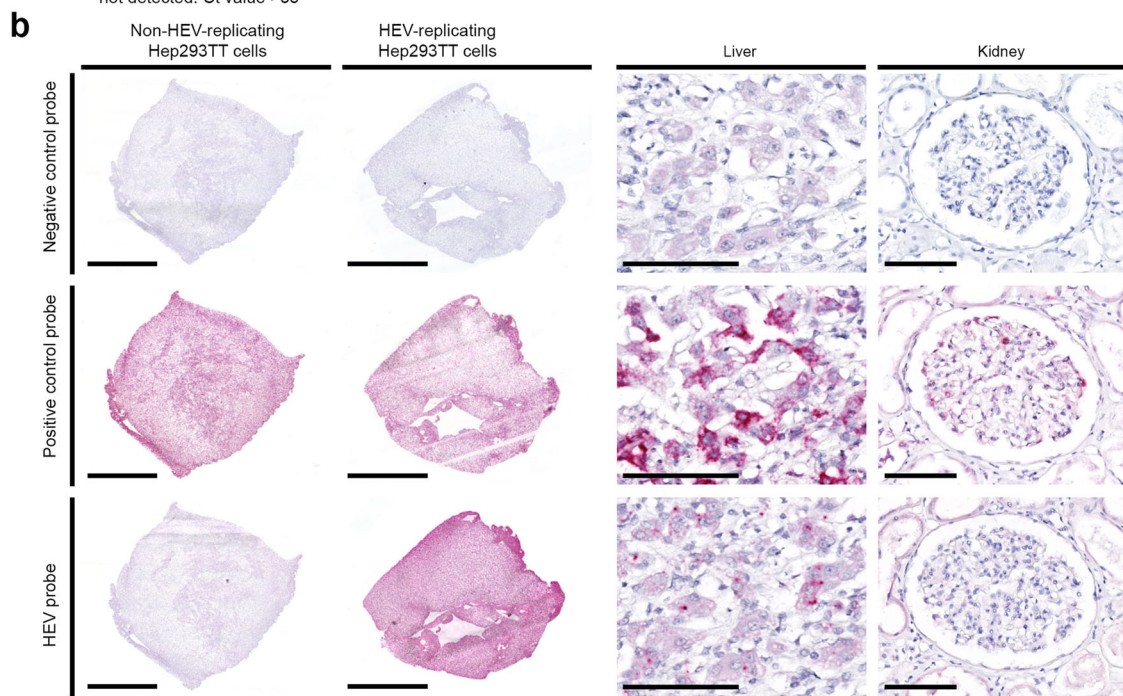

**Fig. 2 | HEV replication detected in the liver but not in the kidney. a** Viral load using RT-PCR and corresponding semi-quantitative IHC results using 1E6 antibody on FFPE sections. **b** RNA in situ hybridization using Dap-B negative control (upper panel), PPIB positive control (middle panel) and HEV-specific (lower panel) probes, on non-HEV-replicating Hep293TT cells and HEV-replicating Hep293TT cells[40] as well as autopsy liver and kidney transplant tissue (Patient 1). Scale bars: 2 mm; liver: 100 μm; kidney: 100 μm (n = 3 experiments, two sections stained per condition and per experiment). Source data are provided as a Source Data file.

only as virions, but also abundantly as free antigen or empty capsid protein, with an obvious discrepancy between the relatively low levels of HEV RNA compared to high levels of HEV ORF2 protein in the urine[9,26]. HEV ORF2 protein trapped in the glomeruli potentially explains the latency observed between viral clearance and restitution of kidney function[12].

Considering the genetic organization and the HEV life cycle, it is not surprising that the HEV ORF2 protein emerges as the key molecule causing extrahepatic manifestations. Unlike HEV ORF1 and ORF3

proteins, the HEV ORF2 protein is produced and secreted in significant excess into the bloodstream, where it exists also in a free form, i.e., not associated with virus particles[18], remarkably stable and prone to precipitate[17,33]. As such, it constitutes the virus main immunogenic structure and a potential immunologic decoy[19,20]. We showed that the glomerular HEV ORF2 protein does not correspond to the infectious form as it is not associated with HEV RNA but displays the molecular weight of a truncated non-glycosylated HEV ORF2 protein, similar as described in the urine and the stool[35,36]. Proteases, as hallmarks of liver

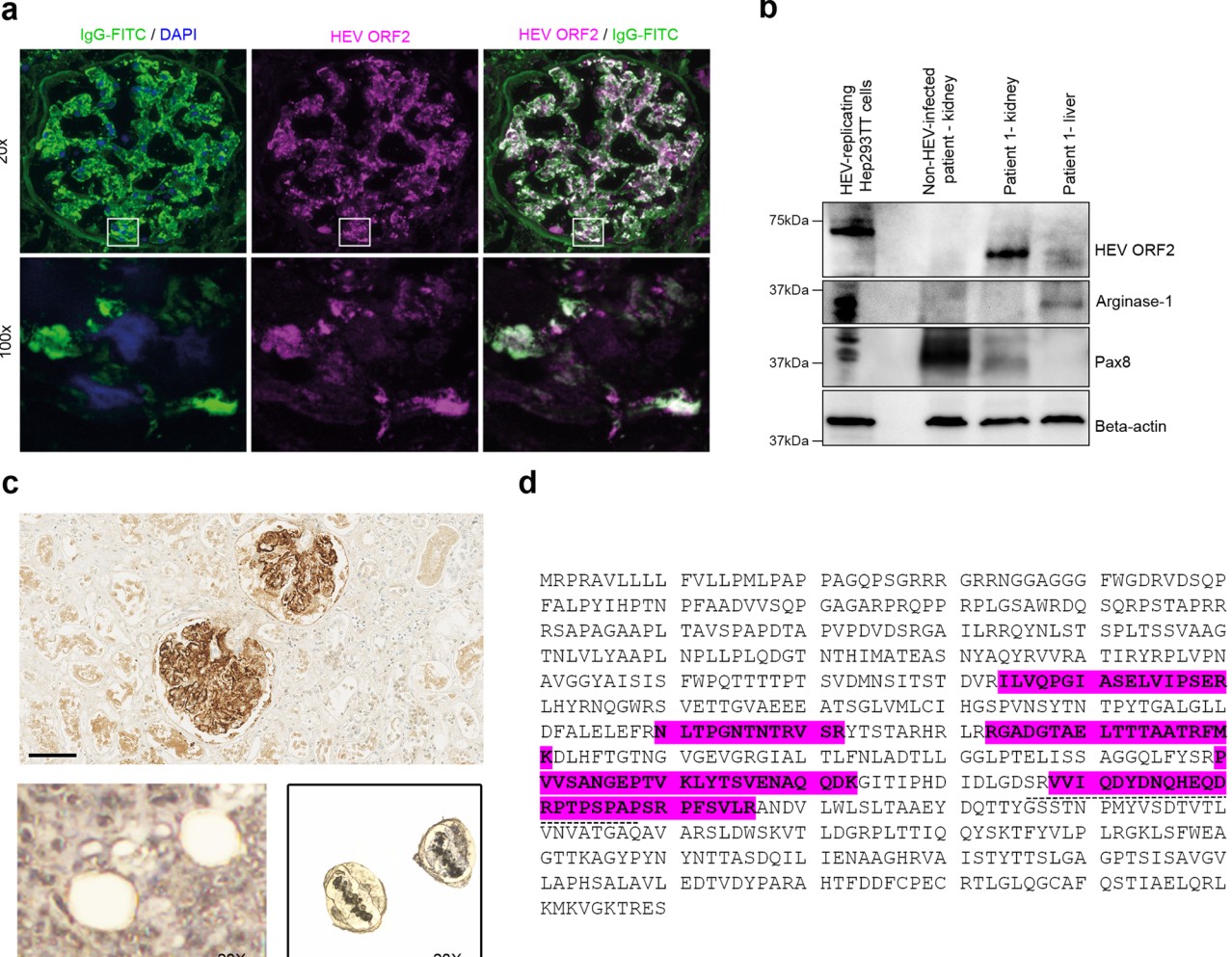

**Fig. 3 | Glomerular deposition of a 60 kDa HEV ORF2 protein in a kidney transplant recipient with hepatitis E. a** Visualization by immunofluorescence staining of a glomerulus from the autopsy transplant kidney (Patient 1) with IgG (left: green, FITC stain; DAPI counter-stain, blue) highlighting the co-localization with HEV ORF2 protein (middle: magenta, Alexa546 stain; right: white indicating co-localization). Overview at low magnification (top row, scale bar: 50 µm, 20x) and high resolution images (bottom row, scale bar: 5 µm, 100x) corresponding to the areas indicated by the white boxes. Highly significant IgG/HEV ORF2 protein co-localization was found on the scale of entire glomeruli (20x, Pearson's correlation coefficient PPC = 0.838 ± 0.039; mean ± s.d., $n = 25$ glomeruli; $p < 10E-10$) as well as for small imaging fields (45–85 µm side length) acquired at high resolution (100x, Zeiss ApoTome; Pearson's coefficient 0.668 ± 0.138, $n = 16$; $p < 10E-10$). For further glomeruli, please see Supplementary Fig. 2b. **b** Western blot analysis for HEV ORF2 protein, the liver marker Arginase-1, and kidney marker Pax8 in HEV-replicating Hep293TT cells, kidney tissue from a non-HEV-infected patient and kidney transplant of Patient 1 (Patient 1 - kidney), as well as liver tissue from Patient 1 (Patient 1 - liver)($n = 1$ blot). **c** Laser-capturing of glomeruli positively stained for HEV ORF2 protein by IHC using 1E6 antibody. Scale bar: 100 µm. After excision (lower panel, 20x). Laser-captured glomeruli in the LCM cap (20x) (50 glomeruli per sample, $n = 2$ samples sourced from 1 stained section). **d** Mass spectrometry analysis of the HEV ORF2 sequence derived from laser-captured glomeruli from the transplant kidney tissue of Patient 1. Glomerular fragments of HEV ORF2 protein highlighted in magenta. The dashed line depicts the epitope of the 1E6 antibody. Source data are provided as a Source Data file.

diseases[37] and various inflammatory glomerular diseases[38] may contribute to the cleavage of the HEV ORF2 protein. This truncated isoform was present in the infected liver (Fig. 3b), reflecting either the passage of this form from the blood and/or the ongoing cleavage by hepatic proteases of the non-glycosylated form, dominantly present in the cytosolic compartment of infected cells[18,19]. This truncated intracellular form is then passively released into the circulation after hepatocyte apoptosis[39].

HEV ORF2 protein IHC and IF, corroborated by WB and MS, allowed us to connect a morphologically variegated and in early stages subtle pattern of glomerular injury (within the spectrum of MPGN) with a specific etiology. Implementation of HEV ORF2 protein immunostaining for routine diagnostics is straightforward[23,40] and allows to delineate hepatitis E-associated GN from GN of other causes. This approach is in accordance with the proposed etiology-based classification[41] and

defines hepatitis E-associated GN as a distinct entity[12]. HEV ORF2 protein immunostaining should be helpful especially in cases in which diagnosis is hampered by limitations of serological testing[1], and in unsuspected cases of hepatitis E in which extrahepatic manifestations predominate. It may guide therapeutic decisions, especially with regard to immunosuppressive treatment and/or antiviral therapy[42].

In summary, the discovery of glomerular IgG/HEV ORF2 protein immune complexes may provide a mechanistic explanation for how this antigen triggers renal disease, in particular immune complex GN. Our findings potentially explain several still incompletely understood observations on hepatitis E-associated renal disease, and establish a molecular link between HEV infection and kidney dysfunction[8]. Finally, they propose HEV ORF2 protein immunostaining as a diagnostic tool for hepatitis E-associated GN, especially in IC-MPGN of clinically unclear origin.

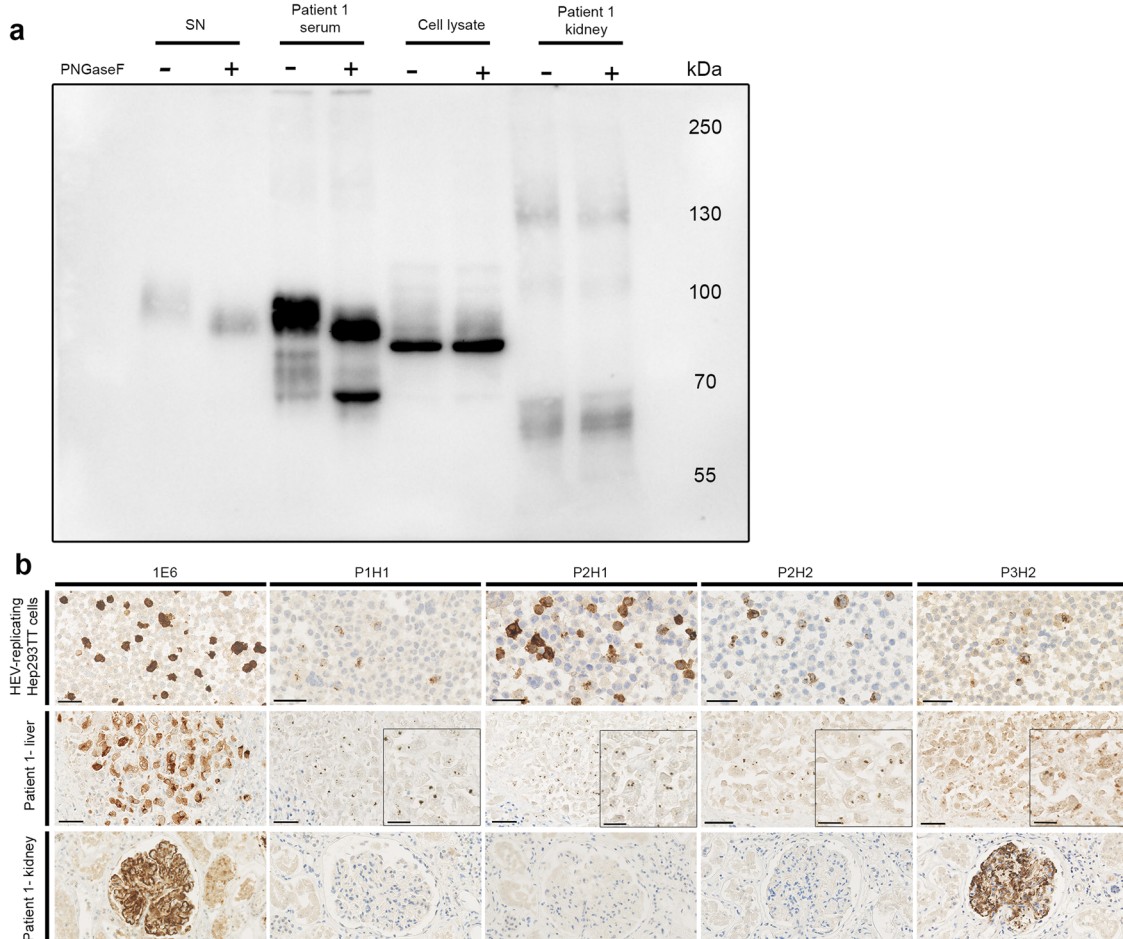

**Fig. 4 | Characterisation of the glomerular HEV ORF2 protein. a** Western Blot with 1E6 antibody combined with PNGase F treatment showing the presence of glycosylated HEV ORF2 protein (HEV ORF2g, 90 kDa) in supernatant of HEV-replicating Hep293TT cells (SN) and in the serum of Patient 1. No evidence of glycosylated protein in the kidney protein extract of Patient 1 (60 kDa bands), as observed in cell lysate of HEV-replicating Hep293TT cells containing intracellular non-glycosylated HEV ORF2intra (72 kDa bands). Detection of bands with higher molecular weight in kidney protein extract of Patient 1 likely corresponding to the immune complexes of HEV ORF2 proteins and immunoglobulins G ($n = 2$ blots). **b** Immunohistochemistry using 1E6 and P3H2 antibodies recognizing all the isoforms of HEV as well as P1H1, P2H1 and P2H2 mAbs recognizing only infectious HEV ORF2i. HEV-replicating Hep293TT cells, as well as liver (Patient 1 - liver) and kidney transplant tissue (Patient 1 - kidney). Scale bars: 50 μm ($n = 3$ experiments per tissue type, each condition in duplicate). Source data are provided as a Source Data file.

## Methods

### Patients and tissue samples
Written informed consent (provided by Springer Nature) was given by the next of kin of all patients, according to CARE guidelines and in compliance with the Declaration of Helsinki principles. It covers the permission to analyse and publish the obtained data, and explicitly states that full anonymity can not be guaranted. Clinical data have been anonymised. Sex and gender were not taken into considerations.

This study was approved by the internal review board of the University Hospital Zurich and the Cantonal Ethics Committee of Zurich, Switzerland (KEK-ZH-Nr. 2013-0504).

Patients' clinical presentation and histopathologic findings are detailed in the Supplementary appendix.

### Histopathologic evaluation of tissue samples
Formalin-fixed, paraffin-embedded (FFPE) liver and kidney specimens were processed according to standard histologic methods including hematoxylin & eosin stain (H&E), liver specimens additionally with periodic acid-Schiff after diastase digestion stain (PAS-D) as well as a connective tissue stain such as Masson trichrome, kidney specimens additionally with periodic acid-Schiff (PAS), silver methenamine stain, elastic van Gieson-stain (EVG) and acid fuchsin orange G stain (AFOG).

Duodenum specimen was processed with Alcian Blue for PAS, in conjunction with PAS staining (AB/PAS) for visualization of the glycocalix.

Histologic slides including archived and newly stained slides from prior kidney biopsies and tissue gathered at autopsy were evaluated independently by experienced renal (BH, AG, HH, HY) or liver (DL, AW) pathologists, respectively.

### Immunohistochemistry (IHC)
Distinct mouse monoclonal antibodies (mAbs) were used to target the different HEV ORF2 protein isoforms: 1E6[40] (Millipore Corporation, MAB8002, clone 1E6) and P3H2[22] for glycosylated (ORF2g/c) and nonglycosylated (ORF2i) isoforms, P1H1, P2H1 and P2H2 for non-glycosylated infectious HEV ORF2i only[18,22]. IgG isotype control antibody (Invitrogen) was used in parallel of HEV ORF2 protein staining using 1E6 antibody according to the standard procedures at the Department of Pathology and Molecular Pathology. FFPE cytospin material from Hep293TT cells replicating a full-length HEV genome HEV or the green fluorescent protein were used as positive and negative controls[40]. Visualization of the HEV ORF2 capsid protein was achieved using 1E6 antibody (as described in ref. 40) and P1H1, P2H1, P2H2 or P3H2 antibodies at a 1/250 dilution including a 90 min pre-treatment at 100 °C with buffer CC1 and direct detection with OptiView

**Table 1 | Clinical findings, laboratory values and HEV ORF2 IHC on autopsy kidneys, patients 1-4[24]**

| | Chronic hepatitis E | Acute hepatitis E | | |
|---|---|---|---|---|
| | Patient 1 | Patient 2 | Patient 3 | Patient 4 |
| HEV genotype | HEV-3h_s | HEV-3h_s | HEV-3 | HEV-3h_s |
| Age: 63 ± 5 years Sex: 3 males, 1 female | 50–59 | 50–59 | 60–69 | 70–79 |
| History of liver disease | chronic HEV infection | cirrhosis due to NASH | cirrhosis due to NASH/ASH | cirrhosis due to ASH |
| Viral load in liver | $4.4 \times 10^6$ | n.a. | n.a | n.a |
| History of kidney disease | IgA nephropathy, kidney TPL | unknown | unknown | unknown |
| Viral load in kidney | 0 | n.a. | n.a. | n.a |
| Immunosuppression (IS) | yes | no | no | no |
| basis IS / add on IS | tacrolimus, MMF / corticosteroids | - | - | - |
| Urea max. (mmol/l) | hemodialysis | n.a. | 25 | n.a. |
| Creatinine max. (umol/l) | 573 | 188 | 180 | > 200 |
| eGFR min. (ml/min/1.73m²) | hemodialysis | 33 | 25 | n.a. |
| Serum albumin min. (g/l) | 11 | 26 | 26 | 24 |
| Proteinuria | yes | yes | yes | n.a. |
| HEV RNA in blood | $1.2 \times 10^8$ IU/mL | $4.0 \times 10^6$ IU/mL | $4.6 \times 10^4$ IU/mL | $2.2 \times 10^3$ IU/mL |
| HEV ORF2 immunohistochemistry on autopsy kidney (scale bar: 50 µm) |  |  |  |  |
| Staining intensity, semiquantitative | +++ | + | + | + |

ASH, alcoholic steatohepatitis; HEV, hepatitis E virus; MMF, mycophenolate mofetil; NASH, nonalcoholic steatohepatitis; TPL, transplantation. Viral load expressed as HEV genome copy number/µg total RNA. n.a., information not available. Scale bars: 50 µm.

Kit (Ventana). For further immunohistochemistry on FFPE sections, standard procedures at the Department of Pathology and Molecular Pathology were applied using the Ventana BenchMark automated staining system with IgG, IgA, IgM, and C3 antibodies (Dako, A0424, P0216, A0426 and A0062, dilution: 1:45 000, 1:30 000, 1:7 500 and 1:2 000 respectively).

## On-slide deglycosylation
For the on-slide deglycosylation experiments, the protocol was adapted from Wang et al.[43]. The sections were deparaffinized, processed for antigen retrieval step (as required for the IHC protocol using 1E6 antibody[40]) and incubated with a deglycosylation mix composed of: protein deglycosylation mix II, beta 1-3 galactosidase and alpha 1-3, 4, 6 galactosidase (New England Biolabs (NEB), P6044Sm P0726S and P0747S respectively), at RT for 30 minutes followed by a 16 h incubation at 37 °C. The sections were then further stained with 1E6 antibody as described in ref. [40]. As control, FFPE sections of human duodenum were stained with AB/PAS preceded or not by an on-slide deglycosylation step to assess the effect on the glycolalyx.

## Immunofluorescence
For IgG/HEV ORF2 co-staining, we used either fresh frozen (6 μm thick) or FFPE (2 μm thick) tissue sections from kidney transplant tissue. Sections were mounted on glass slides (SuperFrost Gold) and dried for 30 min at 37 °C.

Fresh frozen sections were fixed in chemically pure acetone for 10 minutes. Slides were then air dried, pretreated for 5 minutes with a solution of 0.1% polyoxyethylene (20) sorbitan monolaurate (Tween 20) in tris-buffered-saline and washed with distilled $H_2O$. FFPE tissue section were deparaffinized and pretreated with Tris/EDTA/Borat Buffer pH 9.0 for 30 minutes at 100 °C.

Automatic staining was performed on a Leica Bond RX platform. Mouse monoclonal antibody clone 1E6 against the HEV ORF2 protein was incubated for 1 h at a dilution of 1:125 followed by a mix of Alexa Fluor 546-conjugated goat anti-mouse antibody (Invitrogen BV, A11018) and FITC-conjugated Rabbit anti-Human IgG (Gamma chain, Diagnostic Biosystem, F008) for 1 h at a dilution of 1:50. Following automated staining, the slides were hand washed in distilled $H_2O$. Tissue was covered with Vectashield® Antifade Mounting Medium with DAPI (Vector Laboratories, H-1200), covered with a coverslip and stored at 4 °C until evaluation.

For single immunofluorescence on fresh frozen sections, standard procedures at the Department of Pathology and Molecular Pathology were applied using the Leica Bond automated staining system with IgG, IgA, IgM, kappa and lambda, C3, C4d antibodies (Diagnostic Biosystem, F008, F007, F009, F001 and F002, F003, BI-RC4D respectively, dilution: 1:25 for all antibodies).

## Transmission electron microscopy (EM)
Archived images from electron microscopy performed on the allograft kidney biopsies of Patient 1, taken 11 years and 3 months (biopsy 3) as well as 11 years and 4 months post-transplantation (post-TPL) (biopsy 5) were re-evaluated. Additional ultra-thin sections from the archived grids were cut, stained and analyzed in a HITACHI TEM type H-7650 electron microscope.

Tissue samples of the allograft kidney from Patient 1 were collected at autopsy and were fixed in 2.5% buffered glutaraldehyde for at least 12 h, post-fixed for 2 h in 1% osmium tetroxide, dehydrated in a series of graded ethanol solutions and propylene oxide and embedded in epoxy resin 48 h at 60 °C. For comparison purposes, HEV negative tissue samples from the biopsy 1 (1 month post-TPL) of Patient 1 were sourced from archive FFPE material.

Semi-thin sections (1 μm) were stained with methylene blue – azure. Ultra-thin sections (90 nm) were stained with UranyLess and

lead citrate. Sections were examined in a HITACHI TEM type H-7650 electron microscope.

## In situ hybridization (ISH) for HEV RNA
In situ hybridization for HEV RNA was performed as described in ref. [40] using a commercially available probe designated V-HEV, targeting nucleotides 19-7257 of HEV-1 to -4 (Advanced Cell Diagnostics, #468111), a human negative (Dap-B) and a positive control (PPIB) −probes (Advanced Cell Diagnostics, #310043 and #313901 respectively). PPIB gene is expressed at low level in most of the tissues, between 10 to 30 copies per cell, providing a strict control for sample quality and technical performance.

## Laser capture microscopy (LCM)
To perform mass spectrometry analysis of glomerular extracts from Patient 1, glomeruli stained for HEV ORF2 protein (1E6 antibody) were dissected using the ArcturusXT™ LCM System (Thermo Scientific). Protein extracts from one non-HEV-infected human kidney were used as control.

Serial cuts of the formalin-fixed paraffin-embedded kidney specimen from Patient 1 were prepared for LCM: one 2 μm thick section stained using 1E6 antibody, two 10 μm thick sections stained using Cresyl violet placed on polyethylene naphthalate (PEN) membrane glass slide and kept overnight in 4 °C[44,45]. Areas of interest were identified by overlapping the HEV ORF2+ glomeruli with the matching structures on the 10 μm thick section on PEN membrane glass slides. Two 10 μm thick sections from the non-HEV infected human kidney patient were similarly processed for glomeruli isolation. Collection of glomeruli was verified by microscopic examination of the LCM cap as well as the excised region (Fig. 3c). Two caps containing 40-50 glomeruli each were collected per case (one cap per 10 μm thick section, 2 sections per sample type, 4 caps in total). One cap containing interstitial tissue was also prepared per sample type (kidney from Patient 1 and the non-HEV infected patient, 2 caps in total). After excision, the caps containing tissue were transferred to a 1.5 mL centrifuge tube (Eppendorf ®Safe-Lock tubes) and frozen at −20 °C until further processing. As additional controls, liver samples from Patient 1 and non-HEV infected patient were similarly prepared.

## Laser-captured sample preparation for mass spectrometry-based protein identification
For protein extraction, sterile blades and forceps were used to peel off the thermoplastic membranes containing laser-captured glomeruli or interstitial tissue from the LCM caps, which were then transferred into a sterile Eppendorf® Safe-Lock tube.

Eight samples in total ($n = 4$ for laser-captured glomeruli, $n = 2$ for laser-captured interstitial tissue, $n = 2$ for laser-captured liver tissue, from Patient 1 and non-HEV infected patient) were analysed at the Functional Genomic Center Zurich for mass spectrometry (MS) analysis. Briefly, the samples were lysed by adding 4% SDS lysis buffer and incubated for 60 minutes at 95 °C, 1 minute HIFU (High Intensity Focused Ultrasound), 10 minutes sonication, 30 minutes of additional heating at 95 °C, HIFU, sonication and max spin for 5 minutes, and then digested using trypsin solution on a thermo shaker at 37 °C overnight. The digested samples were dried and dissolved in 3% aqueous acetonitrile and 0.1% formic acid before being transferred in vials for liquid chromatography-mass spectrometry analysis. For the kidney samples, 200 ng peptides were injected on a M-class UPLC coupled to an Exploris Orbitrap mass spectrometer (ThermoFisher). The liver samples were loaded on evotips (Evosep) following the provided instructions and analyzed on an Evosep One LC coupled to a TIMS TOF Pro mass spectrometer (Bruker). The acquisition was done in diaPASEF mode.

The acquisition method used contained an inclusion list for the protein of interest, HEV ORF2 protein. For the kidney samples, the

acquired MS data were processed for identification using the Max-quant search engine (V2.0.1.0). The spectra were searched against the HEV ORF2 protein sequence merged with the Homo sapiens protein database Swissprot, and also against interstitial markers (cytokeratin 7, complement 3, and collagen 1A1) and specific glomerular marker podocin[46,47]. For the liver samples, the acquired data independent acquisition spectra were processed with DIA-NN (Version 16[48],) using a library free approach with the same protein database as for Maxquant. Results were summarized in the Proteome Software Scaffold (V 5.0.1) under very stringent settings (1% protein false discovery rate [FDR], a minimum of 2 peptides per protein, 0.1% peptide FDR) and expressed as averaged total spectrum counts ± s.d. (Supplementary Table 2).

## Image processing

Slides with histologic / histochemical stains, immunohistochemistry and in situ hybridization were evaluated by light microscopy and scanned with a resolution of 0.250 μm per pixel (on a Hamamatsu Nanozoomer 2.0 HT whole slide imager) for evaluation by digital microscopy (NDP.view 2 viewer software, Hamamatsu Photonics K.K., V2.7.52). Regions of interest were exported as tif. files from scanned slides, from images acquired from a Zeiss Axio Scope.A1 equipped with a Zeiss Axiocam 100 color camera controlled via ZEN 2.3 lite Software, or from images acquired with an Olympus BX43 microscope equipped with an Olympus DP-43 digital camera controlled via Olympus cellSens software.

Immunofluorescence images were acquired with an upright fluorescence microscope (AxioImager.Z2 controlled by ZEN Blue software; 89 North Photofluor LM-75 light source, and Axiocam 503 mono camera; Zeiss, Jena, Germany), equipped with the following objectives: 20x (NA 0.5, Plan-NEOFLUAR), 40x (NA 1.4 oil, Plan-APOCHROMAT), and 100x (NA 1.45 oil, Plan-APOCHROMAT) objectives. This setup provides an excellent spatial resolution (nominally about 200 nm lateral resolution in our study; pixel size was 45.4 nm for 100x objective) comparable to confocal microscopy[49]. High resolution images were taken with the 100x objective using the ApoTome.2 module with deconvolution (grid 5 lp/mm; section thickness 0.7 μm). We used Vysis Abbott Chroma filter sets (Blue: excitation (ex) 335–383 nm, emission (em) 420–470; green: ex 481–507 nm; em 521–551 nm; red: ex 534–556 nm, em 574–606 nm). Control experiments with separate single-channel IgG or HEV ORF2 staining revealed negligible cross-talk between green and red fluorescence channels.

Co-localization of IgG and HEV ORF2 was quantified using Fiji[50] and the JACoP ImageJ plug-in[51]. For analysis of entire glomeruli, images taken with the 20x objective were cropped to a rectangle around the glomeruli's outlines. For the 100x images, subareas within individual glomeruli of 45-85 μm side lengths were selected. Pearson's correlation coefficient (PCC) was calculated using Costes approach for automatic thresholding[52,53]. To test for significance of PCCs we used Costes randomization (1000 rounds) on each image pair with block sizes of 4 pixel (0.908 μm) for 20x and 10 pixel (0.454 μm) for 100x. *P*-values were all highly significant ($p < 10E-10$). We also calculated the Manders coefficients M1 and M2[54], which for Patient 1 were between 0.64 and 0.98 for 20x images (M1 = 0.84 ± 0.08, M2 = 0.83 ± 0.09; mean ± s.d., $n = 25$ glomeruli) and between 0.22 and 0.83 for 100x images (M1 = 0.58 ± 0.14, M2 = 0.72 ± 0.12; mean ± s.d.), consistent with high degrees of co-localization. Thresholds for M1 and M2 analysis were set manually.

For ultrastructural evaluation, ultrathin sections were examined using a transmission electron microscope (HITACHI H-7650). Regions of interest were exported in tif format. Figures were created using Adobe Photoshop software.

## Western blotting

Autopsy kidney transplant and liver tissue samples from Patient 1 and normal kidney tissue sample from non-HEV-infected patient were homogenized in RIPA buffer (Sigma-Aldrich, R0278-50ML) containing phosphatase inhibitor and protease inhibitor cocktail tablets. Protein extracts were separated by SDS-PAGE on a 4-20% precast poly-acrylamide gel (BioRad, #4561094) transferred to a PVDF membrane according to the manufacturer's protocol and analyzed by Western blotting. The antibodies used were directed against HEV ORF2 protein (Millipore Corporation, MAB8002), PAX8 (Proteintech,10336-1-AP), Arginase-1 (Cell Marque, SP156), and β-Actin (Cell Signaling, 4970 T/S). Protein expression was detected by chemiluminescence and gel-imaging system (Bio Rad).

For PNGase F treatment, samples were denatured for 10 minutes at 95 °C in glycoprotein denaturing buffer (New England Biolabs, NEB). Digestion with Peptide-N-Glycosidase F (PNGaseF, NEB, P0709,) was carried out for 4 h at 37 °C in the presence of 1% NP40 and the buffer provided by the manufacturer (NEB). Samples prepared without gly-cosidase but otherwise under the same conditions were used as controls.

Uncropped blots are provided in the Source Data file.

## Molecular testing: qRT-PCR for HEV

For HEV RNA detection, a qRT-PCR protocol was applied to fresh fro-zen and FFPE tissue specimens. RNA was extracted with the Maxwell RSC simply RNA Kit (for frozen samples) and the Maxwell RSC RNA FFPE kit (for FFPE samples)(Promega) on a Maxwell instrument (Pro-mega) according to the manufacturer's recommendations.

cDNA reverse transcription was carried out using High Capacity cDNA Reverse Transcription Kit (Thermo Fischer) and following manufacturer's recommendations. Briefly, reaction was done in 20 μL using random hexamer and 100 ng of total RNA as template. The cDNA samples were diluted twice for subsequent real-time PCR.

Real-time PCR has been performed as described in Fraga et al.[55] using 2x TaqMan® Universal Master Mix II no UNG (Thermo Fischer) and a Quantstudio 5 apparatus (Thermo Fischer). Each 20-μL PCR reaction was performed with 5 μL of diluted cDNA. A standard curve prepared with dilutions of pUC-HEV-83-2-27 plasmid[56] served to determine HEV genome copy number detected in each tissue sample.

A sample was considered as HEV positive if the average Ct value of the HEV amplicon was below the average Ct value of 35, corresponding to $10^2$ copy/mL.

## Molecular testing: HEV genotyping

Plasma sample was subjected to HEV genotyping following amplifica-tion of HEV ORF2 (592 bp) sequence, as described[55]. Genotype and subtype assignment were performed using the HEVnet genotyping tool (https://www.rivm.nl/mpf/typingtool/hev/).

## Serological assay for anti-HEV IgG and IgM

Serum sample obtained in Cantonal Hospital Aarau was subjected to VIDAS® ANTI-HEV IgG and IgM assay (BioMérieux, France). Sample from University Zürich Hospital was assessed using Enzyme Immu-noAssays for the determination of IgG and IgM antibodies to hepatitis E Virus provided by Dia. Pro Diagnostic Bioprobes Srl (Italy).

## Reporting summary

Further information on research design is available in the Nature Portfolio Reporting Summary linked to this article.

# Data availability

Additional clinical data and original micrographs are available under restricted access for ethical reason and access can be obtained upon request to the corresponding author (AW), within no later than one calendar month. Access will be granted to academic scientists, actively involved in hepatitis E research. Source data for all other relevant raw data are provided in the Source Data file. Image files for the analysis of the co-localisation of HEV ORF2 and IgG within glomeruli have been

deposited to the Zenodo repository at https://doi.org/10.5281/zenodo.13740082[57]. The mass spectrometry proteomics data have been deposited to the ProteomeXchange Consortium via the PRIDE partner repository[58] with the dataset identifier PXD055348. Source data are provided with this paper.

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

## Acknowledgements

We thank for financial support the Swiss National Science Foundation (CRSK-3_190706 to JG and 310030_207477 to DM) as well as the Uni-scientia Stiftung, Zurich and the University Hospital Zurich ("USZ Innovations-Pool")(both to AW). We would like to thank Angela Broggini, PhD, Rita Bopp, André Fritsche, Christiane Mittmann, Carmen Gavrisan, Marcel Glönkler, Doris Guntersweiler and Michael Reinehr, MD for their excellent assistance by performing the autopsy, immune stains and electron microscopy analyses and providing reagents, respectively. The authors gratefully acknowledge the Functional Genomics Center Zurich (FGCZ) of University of Zurich and ETH Zurich, and in particular Dr. Sybille Pfammatter, for the support on proteomics analyses.

## Author contributions

B.H., A.G., and A.W. conceived the original idea. AL.L., B.H., A.G., and A.W. conceived and designed the analysis, collected data, performed the analysis, and wrote the paper. D.L., H.Y., N.S., H.S., H.H., B.M., and D.M. performed the analysis. M.A., J.J., F.H., R.P., M.E.H., A.P., E.M., M.F., L.C., and J.G. contributed data and analysis tools. M.A., J.G., D.M., M.F., and L.C. provided critical feedback and helped shape the manuscript.

## Competing interests

The authors declare no competing interest.

## Additional information

[1]Department of Pathology and Molecular Pathology, University of Zurich (UZH) and University Hospital Zurich (USZ), Zurich, Switzerland. [2]Division of Gastroenterology and Hepatology, Lausanne University Hospital and University of Lausanne, Lausanne, Switzerland. [3]Brain Research Institute, University of Zurich, Zurich, Switzerland. [4]Institute of Pathology, Cantonal Hospital, Aarau, Switzerland. [5]Institute of Veterinary Pharmacology and Toxicology, University of Zurich - Vetsuisse Faculty, Zürich, Switzerland. [6]Department of Visceral Surgery and Medicine, Inselspital, Bern University Hospital, University of Bern, Bern, Switzerland. [7]Univ. Lille, CNRS, Inserm, CHU Lille, Institut Pasteur de Lille, U1019 – UMR 9017 - CIIL - Center for Infection and Immunity of Lille, Lille, France. [8]Clinic of Nephrology, University Hospital Zurich, Zurich, Switzerland. [9]Institute of Medical Genetics and Pathology, University Hospital Basel, University of Basel, Basel, Switzerland. [10]Department of Gastroenterology and Hepatology, University Hospital Zurich, Zurich, Switzerland. [11]Institute of Molecular Cancer Research (IMCR), University of Zurich (UZH), Zurich, Switzerland. [12]These authors contributed equally: Anne-Laure Leblond, Birgit Helmchen. ✉e-mail: achim.weber@usz.ch

