## [Peer Review File · Nature Communications]

Anti-HEV antibody-ORF2 complex deposits are associated with glomerulonephritis in hepatitis E with reduced immune statusREVIEWER COMMENTS

Reviewer #1 (Remarks to the Author):

In this article, the authors reported an explanation of the kidney disorders associated with HEV infection. The authors concluded that that deposition of anti-HEV antibodies and non-infectious ORF2 particles is the cause of disease. The study provided new findings; however, more pieces of evidence are needed to confirm the findings.

Major points

1) Figure 1: The authors showed that HEV ORF2 in liver and kidney, but only RNA in liver using in situ RNA hybridization method.

a) What is the sensitivity assay of in situ RNA method?

2) Lines 100-103: HEV RNA was detected by RNA in situ hybridization only in the liver but not in the kidney

transplant, as determined in biopsies performed 4 and 3 months before death and at autopsy (Supplementary Figure 1a), indicating that the glomerular HEV ORF2 protein was not associated with HEV virion. This conclusion is not solid, since it could be less viral particles in kidney which is expected compared to liver.

3) Supple Figure 1c: The authors detected HEV RNA in FF, but not FFPE of kidney, spleen, and heart.

a) Please provide the viral load in these organs.

b) The presence of RNA in these organs either due to viral replication or passage from blood through these organs. Still there is a possibility of HEV replication.

4) Table 1: please provide the viral load in liver and kidney.

5) Table 1: More ORF2 staining with associated with more viral load in blood and the same for figure1. This indicate the ORF2 is correlated with the HEV RNA and probably to the replication.

6) Figure 2: Lines 127-130 :IHC with mAbs P1H1, P2H1 and P2H2, recognizing only infectious HEV ORF2i17 remained negative in the glomeruli, indicating that the glomerular deposits lacked the nonglycosylated, infectious HEV ORF2i (Figure 2e).

Again this could be due to the sensitivity of assay. What about IHC in spleen, heart, and brain of this patient?

7) What about urine samples of these patients? HEV RNA and ORF2 were detected in the urine of HEV infected patients.

8) The viral genotype in this study is 3h, while most studies showed that 3e and 3f are commonly associated with kidney disease. please discuss this point.

In conclusion: The authors provided new finding about deposition of HEV immunocomplexes in the glomeruli and the immune immunocomplexes mainly include non-infectious ORF2 patients. But this does not exclude the viral replication in these organs. As there organs secondary target for the virus, the replication there is expected to be less. The sensitivity of assays provided by the authors could affect the interpretation of results.

Reviewer #2 (Remarks to the Author):

Although the primary site of infection is the liver, chronic hepatitis E virus (HEV) infections are characterised by extrahepatic manifestations. Among these, renal dysfunction is prominent and progression to end-stage renal disease has been associated with HEV infection. In the study by Helmchen et al, the authors observed immune complex-mediated glomerulonephritis (GN) in a kidney transplant recipient with chronic hepatitis E. The authors did not find any evidence of active HEV infection or replication in the kidney, which is underlined by the finding that the glomerular depositions appear to not contain the infectious, nonglycosylated ORF2. The authors speculate that rather a cleaved form of the glycosylated version of ORF2 complexed with IgGs is accumulating in the kidney, which is secreted by infected cells as an immunological decoy. They were able to confirm their observation in three acute HEV patients, albeit with less severity due to the less advanced stage of the disease in these patients. The extrahepatic manifestations of HEV are poorly understood and this study may help to clarify this aspect.

Main comments:

The study is interesting but very descriptive. The glycosylated HEV capsid is known to be secreted from cells and it is not very surprising that it is excreted via the kidneys. It would be interesting to understand why only some chronic HEV patients, but not all, develop GN. Is there an underlying genetic factor or a particular immune status of the patient that may play a role? Due to the small number of patients who actually developed GN (i.e. 1), it is very difficult to draw general conclusions.

The data on the glycosylated form of ORF2 appear inconclusive. In the western blot shown in figure 2B, the ORF2 protein detected in the kidney is actually too small to be the glycosylated form. In fact, the infectious ORF2 should be 72 kDa (as detected in the HEV+ Hep293TT cells) and the glycosylated ORF2 around 90 kDa. The authors write that this could be the result of posttranslational modification but an easy way to prove that this form corresponds indeed to the glycosylated form would be the treatment with PNGase F.

The P3H2 AB used in their IHC recognises all three forms (infectious, glycosylated and cleaved; Bentaleb et al. 2022) and judging from the WB it could be the intracellular cleaved but not the glycosylated ORF2. How could this form end up in the kidney?

Reviewer #3 (Remarks to the Author):

This is a novel study proposing mechanism for HEV related kidney dysfunction. Authors do acknowledge the limitations yet this study provides potential mechanism of HEV related disease and this is important step in current knowledge of the disease. A handful of case reports are available to date and this is important step in current knowledge of the disease. However, whether the temporal presence of ORF 2 should be considered causative is not certain and may not be stated in those terms atleast until more evidence is available. It was only seen in 1 patient and more evidence is needed before it can be inferred as causative factor. Patient 2-4 in this report had hepatorenal syndrome and hence could not support this hypothesis.

Point-to-point reply letter to the reviewers' comments

We would like to thank the three reviewers for their insightful comments and criticisms, which mainly referred to questions of sensitivity of RNA detection in the kidney and the form of HEV ORF2 protein. The questions raised by reviewers 1-3 have prompted several experiments including additional RT-RNA assays to determine sensitivity, protein assays for HEV ORF2 protein glycosylation status as well as additional mass spectrometry. Moreover, we have restructured the manuscript and reorganized figures, resulting in now four main figures instead of two. Collectively, addressing these issues has substantially improved our manuscript.

Reviewer #1 (Remarks to the Author):

In this article, the authors reported an explanation of the kidney disorders associated with HEV infection. The authors concluded that that deposition of anti-HEV antibodies and non-infectious ORF2 particles is the cause of disease. The study provided new findings; however, more pieces of evidence are needed to confirm the findings.

Major points

1) Figure 1: The authors showed that HEV ORF2 in liver and kidney, but only RNA in liver using in situ RNA hybridization method.

a) What is the sensitivity assay of in situ RNA method?

Response:

For the *in situ* RNA hybridization (ISH) method, we used the positive control probe targeting PPIB gene (Advanced Cell Diagnostics, #313901). This gene is expressed at low level in most of the tissues, between 10 to 30 copies per cell, providing a strict control for sample quality and technical performance.

Staining with PPIB positive control probe gave positive results in all tissues, suggesting that all these specimens are suitable for RNA detection (Reply letter Figure 1). In contrast to the liver, no HEV RNA positive cell was detected in the kidney, heart, brain and spleen.

>We have now incorporated information on the sensitivity of the ISH we had applied in the revised manuscript (Methods section, lines 366-372).

Reply letter Figure 1: ISH staining of heart, brain, spleen, liver and kidney tissue sections of Patient 1 for Dap-B negative control probe, PPIB positive control probe and HEV RNA probe. Corresponding areas. Scale bar: 50µm.

2) Lines 100-103: HEV RNA was detected by RNA in situ hybridization only in the liver but not in the kidney transplant, as determined in biopsies performed 4 and 3 months before death and at autopsy (Supplementary Figure 1a), indicating that the glomerular HEV ORF2 protein was not associated with HEV virion. This conclusion is not solid, since it could be less viral particles in kidney which is expected compared to liver.

Response

We thank the reviewer for her/his critical comment and agree that the sensitivity of the RNA-ISH test we used, as with any other test, has to be considered when drawing any conclusions.

However, the conclusion we made, i.e. that there is no evidence of HEV virion, is based on the results obtained with several assays including RT-PCR and ISH. Nevertheless, we are aware that “absence of evidence does not mean evidence of absence”. At this point, we would like to point out that we do not claim that our findings exclude the presence of HEV virion in the kidney. We only stated that we have no evidence of viral particles in the kidney – which is in contrast to the clear detection of HEV RNA in the liver and in contrast to unequivocally detected IgG/HEV ORF2 protein immune complexes in the kidney.

>We now amended accordingly the description of the ISH results in the manuscript and clarified in the text that our data do not exclude the presence of viral particle in the kidney (Results section, lines 124-126).

3) Supple Figure 1c: The authors detected HEV RNA in FF, but not FFPE of kidney, spleen, and heart.

a) Please provide the viral load in these organs.

Response:

We thank the reviewer for her/his valuable comment. As recommended, we performed new RT-PCR experiments to determine the viral load for the different organs using frozen and FFPE materials (Reply letter Table 1) (Methods section, lines 460-475). Importantly, in contrast to our initial method, a standard curve was prepared with dilutions of pUC-HEV-83-2-27 plasmid ¹ to determine HEV genome copy number detected in each tissue sample. Together with this, we also carried out additional semi-quantitative analysis of HEV ORF2 protein detection by IHC in heart, brain and spleen using FFPE materials (Reply letter Figure 2).

Reply letter Figure 2: HEV ORF2 immunohistochemistry for heart, brain, spleen, liver and kidney tissues of Patient 1. Corresponding areas for IgG isotype control and 1E6 antibody from Patient 1. Scale bars: 50 μ m.

We added additional technical controls: IgG isotype controls and autopsy non-infected tissue controls. We tested P1H1, P2H1/H2, P3H2 and 1E6 antibodies. For all antibodies used, no unequivocally specific reactivity for HEV ORF2 protein or HEV ORF2 protein-associated immune complexes was observed in the heart, brain or spleen tissues (Reply letter Figure 1 and new Supplementary Figure 1).

Reply letter Table 1: Viral load determined by RT-PCR using frozen and FFPE tissues of Patient 1. not detected: Ct value>35. Corresponding semi-quantitative IHC results using 1E6 antibody on FFPE sections.

Organ	Specimen	Material	HEV genome copy number/ug total RNA	HEV ORF2 protein detection
Liver	Autopsy	FF	4.4 x 10 ⁶	++
		FFPE	9.1 x 10 ⁶	
Kidney	Autopsy	FF	not detected	++
		FFPE	not detected	
Heart	Autopsy	FF	1.3 x 10 ⁴	-
		FFPE	not detected	
Brain	Autopsy	FF	1.2 x 10 ⁴	-
		FFPE	not detected	
Spleen	Autopsy	FF	2.3 x 10 ⁴	-
		FFPE	not detected	

>We have now replaced the old figure with the novel RNA data supplemented with information on viral load and corresponding semi-quantitative IHC results for HEV ORF2 protein in Figure 2a. IHC pictures for all organs are now displayed in the new Supplementary Figure 1 and associated results are mentioned in the amended manuscript (Results sections, lines 120-121 and 137-138).

b) The presence of RNA in these organs either due to viral replication or passage from blood through these organs. Still there is a possibility of HEV replication.

Response:

The RT-PCR results confirm the presence of HEV RNA in frozen tissues, with a 2-log scale higher level in the liver compared to the other organs. Only in the liver, the ISH (Figure 2b and Reply letter Figure 1) and IHC staining (Figure 1b, new Supplementary Figure 1 and Reply letter Figure 2) corroborated RT-PCR results (Figure 2a and Reply letter Table 1), supporting the viral replication.

For heart, brain and spleen, there is no evidence of viral replication as we did not visualize HEV RNA or HEV ORF2 protein positive cells in these tissues (Reply letter Figures 1 and 2 respectively). The viral load for these organs mirrors the significant viremia of Patient 1 with passage of HEV RNA through the blood (Reply Letter Table 1).

For the kidney sample, while HEV ORF2 protein was strongly detected by IHC (Figure 1b), the viral load was below the detection threshold (Ct value >35), in accordance with absence of HEV RNA detection by ISH (Figure 2b).

>As commented above, we state that we have no convincing evidence of HEV RNA and virus replication in the kidney. We do not postulate that we could rule this out with absolute certainty. We amended the revised manuscript accordingly (Results section, lines 124-126).

4) Table 1: please provide the viral load in liver and kidney.

Response:

>As recommended and mentioned above, we added the viral load for liver and kidney in Table 1 of the revised manuscript (Page 33).

5) Table 1: More ORF2 staining with associated with more viral load in blood and the same for figure1. This indicate the ORF2 is correlated with the HEV RNA and probably to the replication.

Response:

We agree with the reviewer's statement. The results of IHC for HEV ORF2 and RT-PCR/ISH for HEV RNA are concordant in the liver of Patient 1 and corroborate the chronic HEV infection (Reply letter Figure 3).

In the heart, brain and spleen tissues, only the RT-PCR results obtained from the frozen materials showed the presence of the HEV RNA, at lower levels ($\sim 10^4$ HEV genome copy number/ μg total RNA) in these organs compared to the liver ($\sim 10^6$ HEV genome copy number/ μg total RNA), reflecting the transient presence of the HEV RNA within the tissues due to viremia (Figure 2a).

In the kidney tissue, there is no clear evidence of the presence of HEV RNA and viral replication:

- The viral load now expressed as HEV genome copy number/ μg total RNA was too low to be detected by RT-PCR, as compared to the other organs (Figure 2a and Reply letter Table 1),
- the ISH staining with HEV RNA probe remained negative despite the reactivity of PPIB positive control probe on corresponding areas (Figure 2b and Reply Letter Figure 1), and
- the IHC and co-IF staining clearly indicated the extracellular localisation of HEV ORF2 protein within the glomeruli, supporting the hypothesis of protein deposition and not cell infection (Figures 1 and 3, and Supplementary Figure 2b).

>Details of the ISH staining sensitivity and new RT-PCR method are included in the methods section (lines 366-372 and lines 461-476). The extracellular localisation of HEV ORF2 protein within glomeruli is now specified in the description of the results (lines 132-134).

Reply letter Figure 3: Schematic representation of the results of immunohistochemistry, in situ hybridization and virus load by RT-PCR for Patient 1. Semi-quantitative analysis.

Nevertheless, to further address this question, we performed a series of new experiment. As already carried out for the detection of HEV ORF2 protein in the glomeruli, we now endeavoured to identify the HEV ORF1 and ORF3 proteins by mass spectrometry, which are required for HEV replication.² We preferred this approach over immunohistochemistry because of the ultra-low abundancy in primary human tissue of both HEV ORF1 and ORF3 proteins. As we have shown, immunohistochemistry for these proteins is not reliable for visualization in primary human tissues (in contrast to experimental system).³

We then were wondering if we could gain informative value from searching for the presence of HEV ORF1 protein, known to be crucial for HEV replication, and/or HEV ORF3 protein.² Given the tight expression regulations in time and space of HEV ORF1 and ORF3, we amended our protocol of data acquisition^{4, 5} and analysis of mass spectrometry^{6, 7}, considering a positive result from the first peptide identified. Protein extract of the liver of Patient 1 was used as positive control. Non-infected liver and kidney tissues were used as negative controls. Detection of albumin was used as internal technical control. The results are expressed as number of different detected peptides and average abundance (hits) per type of tissues (Reply Letter Table 2).

Reply letter Table 2: Detection of the HEV viral proteins ORF1, ORF2 and ORF3 in the liver and kidney using mass spectrometry. Results expressed as number of different detected peptides (upper lane in bold) and average abundance (number of hits) per type of tissues (lower lane)

	Patient 1		Non-infected patient	
	Liver	Kidney	Liver	Kidney
Number of proteins	5248	5626	5431	5085
HEV pORF1	1 peptide 4.5 x 10 ³	0 peptide	0 peptide	
HEV pORF2	9 peptides 8 x 10 ⁵ 7.5 x 10 ⁵			
HEV pORF3	0 peptide			
Albumin	49 peptides			
	16 x 10 ⁶	12 x 10 ⁶	9 x 10 ⁶	20 x 10 ⁶

Among all the diseased and control samples, only one peptide of HEV ORF1 (4.5 x 10³ hits) was detected in the infected liver but not in the infected kidney, despite its ultra-low abundance.

No resulting peptide of HEV ORF3 protein was found in the tissues using this method. Beside its low abundance, HEV ORF3 is also a very small protein (13kDa), substantially limiting its detection by mass spectrometry.

This approach confirmed the viral replication in the liver, i.e. HEV ORF1 resulting peptide found and did once again not provide evidence of viral replication in the kidney.

>Collectively, these findings further support our data. However, given the ultra-low abundance of these two viral proteins, which brings mass spectrometry to the limits of its informative value for this application, and also taking into account the already convincing data, we decided not to incorporate these findings in our revised manuscript.

6) Figure 2: Lines 127-130: IHC with mAbs P1H1, P2H1 and P2H2, recognizing only infectious HEV ORF2i⁸ remained negative in the glomeruli, indicating that the glomerular deposits lacked the nonglycosylated, infectious HEV ORF2i (Figure 2e). Again this could be due to the sensitivity of assay.

Response:

We thank the reviewer for pointing out the IHC results with P1H1 and P2H1/H2 antibodies. We agree that the lack of sensitivity needs to be considered. We showed that the HEV ORF2 protein is present in the kidney based on the immuno-reactivity using 1E6 antibody, by IHC, co-IF and WB (Figures 1 and 3), and also based on the use of mass spectrometry with the detection of the epitope of 1E6 antibody. The discordance of the IHC results using P1H1, P2H1/H2 antibodies between liver and kidney contrasts with the remarkable immuno-reactivity using 1E6 antibody. Interestingly, this reactivity by WB is more pronounced in the kidney than in the liver, suggesting that HEV ORF2 protein is more abundant in the kidney than in the liver (Figure 3b). However, IHC method using P1H1, P2H1/H2 antibodies in the liver was still sensitive enough to lead to positive reactivity. Based on the stoichiometry of HEV ORF2 protein in the liver and kidney, we can therefore conclude that the negative IHC results with P1H1 and P2H1/H2 are due to the fact that the infectious HEV ORF2 isoform is at least not the dominant form in the kidney or missing at all. IHC results using P1H1, P2H1/H2, corroborate the RT-PCR and ISH results (Reply letter Figure 3), supporting that there is no infection in the kidney cells and this glomerular isoform is not associated with HEV RNA.

>We modified the description of these results in the revised manuscript (Results section, lines 139-140).

What about IHC in spleen, heart, and brain of this patient?

Response:

As described above, we performed IHC experiments in the heart, brain and spleen tissues for HEV ORF2 protein (Supplementary Figure 1 and Reply letter Figure 2). We added additional technical controls: IgG isotype controls and autopsy non-infected tissue controls.

No unequivocally specific reactivity for HEV ORF2 protein or HEV ORF2 protein-associated immune complexes was observed in the heart, brain and spleen tissues regardless of the antibodies used (1E6, P1H1 or P2H1/H2) (Reply letter Figure 2 for 1E6 antibody).

>We now mentioned these data in the revised manuscript (Results sections, lines 120-121 and 137-138).

7) What about urine samples of these patients? HEV RNA and ORF2 were detected in the urine of HEV infected patients.

Response:

We agree with the reviewer that information on HEV ORF2 protein concentration in urine might have high informational value and would therefore be desirable. Unfortunately, patient`s urine

samples were not stored. Naturally, we could not compensate for this omission after the death of the patients. This point is indeed a limitation of our study. However overall, we do not consider it critical since this information does not affect the main observation and conclusion of our study. It is good scientific practice to mention in the manuscript discussion the possibility that renal glomeruli could serve as a reservoir for HEV ORF2 proteins, especially in the light of recently published work.⁹

8) The viral genotype in this study is 3h, while most studies showed that 3e and 3f are commonly associated with kidney disease. please discuss this point.

Response:

We thank the reviewer for raising this point. Indeed, studies conducted in France revealed that most patients with hepatitis E who developed kidney disease were infected with gt 3e or 3f.¹⁰ However, in a more recent study, conducted in Germany, by Choi M. *et al.*, 2018¹¹, most patients suffering from HEV-associated kidney disease were infected with gt 3c. Interestingly, this latter genotype belongs to the same clade, i.e. 3.1, than gt 3h while gt 3e and 3f are from clade 3.2. It is likely that the observed differences are linked to the prevalence of the HEV subtypes in the respective regions. Therefore, it explains that our findings were made following gt 3h infection, as this subtype circulates predominantly in Switzerland. Overall, we think that our findings are likely not specific to gt 3h and that it should be explored in the future in the framework of a multicentric analysis.

In conclusion: The authors provided new finding about deposition of HEV immunocomplexes in the glomeruli and the immune immunocomplexes mainly include non-infectious ORF2 patients. But this does not exclude the viral replication in these organs. As there organs secondary target for the virus, the replication there is expected to be less. The sensitivity of assays provided by the authors could affect the interpretation of results.

Response:

As outlined above, we clarified that the fact that after extensive search we found no evidence of viral replication in the kidney, our data does not definitively exclude the presence of viral particles in the kidney (Results section, lines 124-126).

Reviewer #2 (Remarks to the Author):

Although the primary site of infection is the liver, chronic hepatitis E virus (HEV) infections are characterised by extrahepatic manifestations. Among these, renal dysfunction is prominent and progression to end-stage renal disease has been associated with HEV infection. In the study by Helmchen et al, the authors observed immune complex-mediated glomerulonephritis (GN) in a kidney transplant recipient with chronic hepatitis E. The authors did not find any evidence of active HEV infection or replication in the kidney, which is underlined by the finding that the glomerular depositions appear to not contain the infectious, nonglycosylated ORF2. The authors speculate that rather a cleaved form of the glycosylated version of ORF2 complexed with IgGs is accumulating in the kidney, which is secreted by infected cells as an immunological decoy. They were able to confirm their observation in three acute HEV patients, albeit with less severity due to the less advanced stage of the disease in these patients. The extrahepatic manifestations of HEV are poorly understood and this study may help to clarify this aspect.

Main comments:

The study is interesting but very descriptive. The glycosylated HEV capsid is known to be secreted from cells and it is not very surprising that it is excreted via the kidneys. It would be interesting to understand why only some chronic HEV patients, but not all, develop GN. Is there an underlying genetic factor or a particular immune status of the patient that may play a role? Due to the small number of patients who actually developed GN (i.e. 1), it is very difficult to draw general conclusions.

Response:

We thank the reviewer for this valuable comment asking about candidate factors that are likely to determine whether and to what extent hepatitis E-associated GN develops. Indeed, both factors mentioned by the reviewer, i.e. genetic determinants and the patients' immune status, are strong candidates as GN-determining factors. Naturally, the small cohort of four patients does not allow for a robust, statistically substantiated analysis and therefore only limited conclusions to this specific question can be drawn from our study, in particular with respect to genetic factors. Nevertheless, it would seem obvious that the accumulated amount of HEV ORF2 protein produced during infection is a critical factor. This assumption is in line with the findings in our cohort, in which the patient with chronic hepatitis E, associated with long term, high level HEV ORF2 protein production, developed more severe GN than the three patients with acute hepatitis E. This is directly linked to the patient's immune status, as it is well-

documented that the immune status determines the duration and severity of the hepatitis E.¹² However, it can be expected that the connection between the immune status and the question of the development of a GN is even more complex. It can be assumed that patients who have reduced antibody levels against HEV e.g. as a result of an immune deficiency of any kind (including iatrogenic such as immunosuppressive drugs after transplantation) or dialysis are less likely to form immune complexes.¹³

>In the revised manuscript, we have now expanded the discussion part with respect to genetics, duration of disease and patients` immune status as potential GN-determining factors (lines 233-243).

The data on the glycosylated form of ORF2 appear inconclusive. In the western blot shown in figure 2B, the ORF2 protein detected in the kidney is actually too small to be the glycosylated form. In fact, the infectious ORF2 should be 72 kDa (as detected in the HEV+ Hep293TT cells) and the glycosylated ORF2 around 90 kDa. The authors write that this could be the result of posttranslational modification but an easy way to prove that this form corresponds indeed to the glycosylated form would be the treatment with PNGase F.

Response:

As commented by the reviewer, the glomerular isoform does not resemble any intact HEV ORF2 isoforms in terms of molecular weights. The molecular weight of 72kDa or 90kDa for infectious and glycosylated isoforms (respectively), corresponds to wild type HEV ORF2 proteins isolated in cell culture models or in patient serum¹⁴. Recent findings in stool and urine^{9, 15} highlighted the presence of truncated non-glycosylated HEV ORF2 protein, but little is known about intravital HEV ORF2 protein, within tissues, in particular human kidney with glomerular lesions.

The intact ORF2 protein sequence contains three highly conserved potential N-glycosylation sites, named N1, N2 and N3, located in position 137, 310 and 562 respectively.¹⁶ P2H1/H2 antibodies target the 14 amino-acids covering the glycosylation site N3, if the site is not glycosylated.⁸ Refining the glycosylation status of this isoform would be a means to determine its origin, e.g. if it derives either from the non-glycosylated intracellular/infectious HEV ORF2 protein or from the glycosylated secreted HEV ORF2g/c protein.

Based on the recommendation of the reviewer, we performed further Western blots combined with glycosidase treatment using PNGase F (Figure 4a). The molecular weight of the bands

observed in the kidney samples were unchanged after deglycosylation, meaning that the isoform is not glycosylated due to the fact that either the glycosylation site(s) are not glycosylated or they are absent.

We then took advantage of recently available antibodies: P1H1 that is specific of the N terminal of the non-glycosylated infectious HEV ORF2i protein and P2H1/H2 that target the glycosylation site in position 562 (N3).⁸ Unlike in HEV-replicating cells and the liver tissue, IHC with these antibodies led to negative results in the glomeruli, suggesting that the epitopes of these antibodies are absent (Figure 4b). After on-slide deglycosylation, IHC with P2H1 antibody remained negative, confirming the absence of the N3 glycosylation site (Supplementary Figure 4).

Collectively these findings led to the conclusion that the truncated 60kDa glomerular HEV ORF2 protein is not glycosylated and has lost its N-terminal and the N3 glycosylation site.

Reply Letter Table 3: Deglycosylation experiments on kidney sample

Kidney	1E6 Antibody		P2H1 antibody	
	-	+	-	+
Deglycosylation	-	+	-	+
Results	HEV ORF2 protein detected	No change	HEV ORF2 protein not detected – N3 site is either occupied or absent	No change
Conclusions	Non-glycosylated HEV ORF2		N3 glycosylation site is absent	

Finally, we repeated the PNGase F treatment on the glomerular extracts used for mass spectrometry analysis and found only one resulting peptide of HEV ORF2 protein. This ultra-low protein abundance prevented the possibility of further exploitation of this approach and to reach a definitive conclusion.

>We incorporated novel approaches for deglycosylation experiments (on-slide deglycosylation: lines 320-328; WB and PNGase F treatment: lines 455-459), data (Figure 4) and statements in the revised manuscript (Results section, lines 156-179).

The P3H2 AB used in their IHC recognises all three forms (infectious, glycosylated and cleaved; Bentaleb et al. 2022) and judging from the WB it could be the intracellular cleaved but not the glycosylated ORF2. How could this form end up in the kidney?

Response:

We thank the reviewer for this important comment. Indeed, as discussed above, we have evidence that the glomerular isoform is not glycosylated and does not contain the glycosylation site N3. This truncated form could thus derive from the non-glycosylated intracellular isoform (HEV ORF2_{intra}) as well as from the infectious particles. In case of the latter, following HEV infection and apoptosis of the infected hepatocytes, it is conceivable that this genome-free protein had been released into the circulation¹⁷, accumulated in the glomerular compartment and formed immune complexes.

>We incorporated this statement in the discussion of the revised manuscript (lines 256-265).

Reviewer #3 (Remarks to the Author):

This is a novel study proposing mechanism for HEV related kidney dysfunction. Authors do acknowledge the limitations yet this study provides potential mechanism of HEV related disease and this is important step in current knowledge of the disease. A handful of case reports are available to date and this is important step in current knowledge of the disease.

We thank Reviewer 3 for her/his overall positive feedback considering our work an “important step in current knowledge of the disease”. To her/his specific concern we would like to respond as follows.

However, whether the temporal presence of ORF 2 should be considered causative is not certain and may not be stated in those terms at least until more evidence is available. It was only seen in 1 patient and more evidence is needed before it can be inferred as causative factor.

Response:

We proposed a causal relationship for HEV infection and immune complex glomerulonephritis (IC-GN), based on morphological observations and diagnostic examinations, correlation with the clinical context, laboratory findings and several in depth analyses, among others mass spectrometry and co-IF staining. Infection-related immunopathogenesis is a well-established pathology causing IC-GN in particular for virally induced GN. For some glomerulonephritides, this can be shown in animal models, such as in cryoglobulinemia leading to membranoproliferative GN (MPGN) in mice similar to HCV-associated IC-MPGN in humans. The postulate that HEV is causative for the pathogenesis of GN in the Patient 1 is also, but not solely, based on the close temporal association with the course of the disease. It is supported by several corroborating findings including the co-localization of HEV ORF2 protein with IgG and C3, the ultrastructural findings and mass spectrometry.

Further arguments in favor of a causal - and not just temporal - association are the analogy to HCV infection: HCV is proven to be causative for the development of HCV-associated MPGN. Moreover, the biological plausibility and coherence as well as the finding of a biological gradient are all in favor of a causality. Regarding HEV-associated glomerulonephritis, we consider the results of our study highly relevant in establishing this disease as a distinct entity, similar to other well-characterized infection-associated glomerular diseases. Hopefully, this will raise awareness to this under-recognized disease with its potentially fatal consequences.

>To explain more clearly our argumentation favoring causality, we added the following text on lines 208-243 of the main manuscript (Discussion section):

“Infection-related immunopathogenesis is known as a possible cause of GN. It occurs either via in situ immune complex formation as in poststreptococcal GN, via deposition of circulating immune complexes as e.g. in HCV, HIV and EBV or via direct cytotoxic effects of pathogens as for example in HIV, EBV and SARS-CoV-2.¹⁸ Chronic viral infections, such as HCV and HBV, with or without circulating cryoglobulins, are an important cause of MPGN.¹⁹ In 1993, HCV was recognized as a common cause of immune complex-mediated MPGN.^{20, 21} There is no animal model for a HCV-induced GN, in which the immune complex glomerulonephritis is directly caused by an infection of the experimental animal with the HCV. However, there is a mouse model that closely mimics HCV-induced MPGN. In this model, a systemic inflammation occurs affecting, among other organs, the kidney. These mice consistently develop glomerular disease in form of MPGN that closely resembles that seen in humans.^{22, 23}

We have observed the same type of immune complexes in both acute and chronic hepatitis E, but more pronounced in the latter, in accordance with significantly higher HEV ORF2 protein levels found in sera from chronically as compared to acutely HEV-infected individuals.²⁴ In patients 2-4, renal dysfunction was due to hepato-renal syndrome. Nevertheless, it is conceivable that in addition to cases of fully developed glomerulonephritis^{10, 25-27}, as in patient 1, glomerular damage associated with more subtle HEV-ORF2 protein deposits, as in patients 2-4, may represent a general early morphological correlate and harbinger of impaired glomerular function in the context of HEV infection.²⁸

We suggest a causal relationship for HEV and IC-MPGN due to biological plausibility and coherence: in particular 1) the co-localization of HEV ORF2 protein with IgG and C3 in the glomerular lesions, accompanied by increased proteinuria, and 2) the finding of a biological gradient.²⁹

Based on our observations, we cannot deduce whether HEV ORF2 protein-associated immune-mediated glomerular damage also occurs in acute or subclinical hepatitis E in immunocompetent individuals, in the course of which (transient) impaired renal function has also been described.^{30, 31} However, the host immune status determines the duration of HEV persistence and thus indirectly also the amount of HEV ORF2 protein formed cumulatively.^{24, 32} Several factors potentially determine the development and the severity of GN, including genetic and immunologic on the host side. If the immune status is the decisive factor determining the extent of glomerular damage, it is expected to be lower in acute than in chronic hepatitis E, in line with our observations. However, a larger cohort would be necessary to

investigate the candidate factors that are likely to determine whether and to what extent hepatitis E-associated GN develops.”

Patient 2-4 in this report had hepatorenal syndrome and hence could not support this hypothesis.

Response:

Patients 2-4 had acute hepatitis E and hepatorenal syndrome explained their renal failure. We are not assuming that the glomerular findings in these patients caused renal failure, but that they represent an example for the biological gradient with early, very mild HEV-associated deposits which could progress to clinically evident GN in a chronic course of hepatitis E.

References

- [1] Shiota T, Nishikawa S, Endo T. Analyses of protein-protein interactions by in vivo photocrosslinking in budding yeast. *Methods Mol Biol* **1033**, 207-17 (2013).
- [2] LeDesma R, Nimgaonkar I, Ploss A. Hepatitis E Virus Replication. *Viruses* **11**, (2019).
- [3] Lenggenhager D, Gouttenoire J, Malehmir M, et al. Visualization of hepatitis E virus RNA and proteins in the human liver. *J Hepatol* **67**, 471-9 (2017).
- [4] Hughes CS, Foehr S, Garfield DA, Furlong EE, Steinmetz LM, Krijgsveld J. Ultrasensitive proteome analysis using paramagnetic bead technology. *Mol Syst Biol* **10**, 757 (2014).
- [5] Leutert M, Rodriguez-Mias RA, Fukuda NK, Villen J. R2-P2 rapid-robotic phosphoproteomics enables multidimensional cell signaling studies. *Mol Syst Biol* **15**, e9021 (2019).
- [6] Demichev V, Messner CB, Vernardis SI, Lilley KS, Ralser M. DIA-NN: neural networks and interference correction enable deep proteome coverage in high throughput. *Nat Methods* **17**, 41-4 (2020).
- [7] Panse C, Trachsel C, Turker C. Bridging data management platforms and visualization tools to enable ad-hoc and smart analytics in life sciences. *J Integr Bioinform* **19**, (2022).
- [8] Bentaleb C, Hervouet K, Montpellier C, et al. The endocytic recycling compartment serves as a viral factory for hepatitis E virus. *Cell Mol Life Sci* **79**, 615 (2022).
- [9] Ying D, He Q, Tian W, et al. Urine is a viral antigen reservoir in hepatitis E virus infection. *Hepatology* **77**, 1722-34 (2022).
- [10] Kamar N, Weclawiak H, Guilbeau-Frugier C, et al. Hepatitis E virus and the kidney in solid-organ transplant patients. *Transplantation* **93**, 617-23 (2012).
- [11] Choi M, Hofmann J, Kohler A, et al. Prevalence and Clinical Correlates of Chronic Hepatitis E Infection in German Renal Transplant Recipients With Elevated Liver Enzymes. *Transplant Direct* **4**, e341 (2018).

- [12] European Association for the Study of the Liver. Electronic address eee, European Association for the Study of the L. EASL Clinical Practice Guidelines on hepatitis E virus infection. *J Hepatol* **68**, 1256-71 (2018).
- [13] Krain LJ, Nelson KE, Labrique AB. Host immune status and response to hepatitis E virus infection. *Clin Microbiol Rev* **27**, 139-65 (2014).
- [14] Montpellier C, Wychowski C, Sayed IM, et al. Hepatitis E Virus Lifecycle and Identification of 3 Forms of the ORF2 Capsid Protein. *Gastroenterology* **154**, 211-23 e8 (2018).
- [15] Nishiyama T, Umezawa K, Yamada K, et al. The Capsid (ORF2) Protein of Hepatitis E Virus in Feces Is C-Terminally Truncated. *Pathogens* **11**, 24 (2021).
- [16] Ankavay M, Montpellier C, Sayed IM, et al. New insights into the ORF2 capsid protein, a key player of the hepatitis E virus lifecycle. *Sci Rep* **9**, 6243 (2019).
- [17] Vitale I, Pietrocola F, Guilbaud E, et al. Apoptotic cell death in disease-Current understanding of the NCCD 2023. *Cell Death Differ* **30**, 1097-154 (2023).
- [18] Anders HJ, Kitching AR, Leung N, Romagnani P. Glomerulonephritis: immunopathogenesis and immunotherapy. *Nat Rev Immunol* **23**, 453-71 (2023).
- [19] Sethi S, Fervenza FC. Membranoproliferative glomerulonephritis--a new look at an old entity. *N Engl J Med* **366**, 1119-31 (2012).
- [20] Johnson RJ, Gretch DR, Yamabe H, et al. Membranoproliferative glomerulonephritis associated with hepatitis C virus infection. *N Engl J Med* **328**, 465-70 (1993).
- [21] Doutrlepont JM, Adler M, Willems M, Durez P, Yap SH. Hepatitis C infection and membranoproliferative glomerulonephritis. *Lancet* **341**, 317 (1993).
- [22] Kowalewska J. Cryoglobulinemic glomerulonephritis--lessons from animal models. *Folia Histochem Cytobiol* **49**, 537-46 (2011).
- [23] Taneda S, Segerer S, Hudkins KL, et al. Cryoglobulinemic glomerulonephritis in thymic stromal lymphopoietin transgenic mice. *Am J Pathol* **159**, 2355-69 (2001).

[24] Behrendt P, Bremer B, Todt D, et al. Hepatitis E Virus (HEV) ORF2 Antigen Levels Differentiate Between Acute and Chronic HEV Infection. *J Infect Dis* **214**, 361-8 (2016).

[25] Marion O, Abravanel F, Del Bello A, et al. Hepatitis E virus-associated cryoglobulinemia in solid-organ-transplant recipients. *Liver Int* **38**, 2178-89 (2018).

[26] Pischke S, Hartl J, Pas SD, Lohse AW, Jacobs BC, Van der Eijk AA. Hepatitis E virus: Infection beyond the liver? *J Hepatol* **66**, 1082-95 (2017).

[27] Pischke S, Tamanaei S, Mader M, et al. Lack of Evidence for an Association between Previous HEV Genotype-3 Exposure and Glomerulonephritis in General. *Pathogens* **11**, (2021).

[28] Wallace SJ, Swann R, Donnelly M, et al. Mortality and morbidity of locally acquired hepatitis E in the national Scottish cohort: a multicentre retrospective study. *Aliment Pharmacol Ther* **51**, 974-86 (2020).

[29] Hill AB. The environment and disease: association or causation? 1965. *J R Soc Med* **108**, 32-7 (2015).

[30] Dalton HR, Izopet J. Transmission and Epidemiology of Hepatitis E Virus Genotype 3 and 4 Infections. *Cold Spring Harb Perspect Med* **8**, a032144 (2018).

[31] Lhomme S, Marion O, Abravanel F, Izopet J, Kamar N. Clinical Manifestations, Pathogenesis and Treatment of Hepatitis E Virus Infections. *J Clin Med* **9**, (2020).

[32] Kamar N, Selves J, Mansuy JM, et al. Hepatitis E virus and chronic hepatitis in organ-transplant recipients. *N Engl J Med* **358**, 811-7 (2008).

REVIEWERS' COMMENTS

Reviewer #4 (Remarks to the Author):

The authors have significantly enhanced and deepened their study through the integration of additional results. The analysis employs state-of-the-art techniques and recently developed tools for the detection and characterization of various ORF2 proteins. The identification of truncated, non-glycosylated, and noninfectious ORF2 proteins in the kidney has been conclusively delineated, markedly strengthening the manuscript. This reviewer's comments have been comprehensively addressed.

Reviewer #5 (Remarks to the Author):

I am reviewing this paper for the first time after it has already been reviewed by other reviewers. It is a very serious paper on an important topic and the authors have implemented the reviewers' advice as far as possible, which is why I generally recommend "acceptance". However, I still have a few small requests for changes.

Minor points:

☒ Introduction line 60: „In resource-limited countries, endemic and epidemic HEV-1 and -2 are transmitted from person to person mainly through contaminated drinking water. “ The term “from patient to patient² might be interpreted misleadingly... Please delete these 3 word.

☒ Liene 75-77 this is misleading. It should be mentioned that the observed cases occurred in solid organ transplant recipients but not in immunocompetent individuals

2 Results section: The results section starts directly with the autopsy of the patient. This is unusual. Normally you first write "patient xx years old,sex.... transplanted xx years ago.. underlying disease... immunosuppression....."

Sven Pischke

Response to referees
Latest reviewer comments as per the “author checklist” file
“Remaining reviewer comments”

Reviewer #4: The authors have significantly enhanced and deepened their study through the integration of additional results. The analysis employs state-of-the-art techniques and recently developed tools for the detection and characterization of various ORF2 proteins. The identification of truncated, non-glycosylated, and noninfectious ORF2 proteins in the kidney has been conclusively delineated, markedly strengthening the manuscript. This reviewer's comments have been comprehensively addressed.

> We thank the Reviewer for their positive appreciation of our work.

Reviewer #5: I am reviewing this paper for the first time after it has already been reviewed by other reviewers. It is a very serious paper on an important topic and the authors have implemented the reviewers' advice as far as possible, which is why I generally recommend "acceptance". However, I still have a few small requests for changes.

Minor points:

\ **Introduction line 60: „In resource-limited countries, endemic and epidemic HEV-1 and -2 are transmitted from person to person mainly through contaminated drinking water. “ The term “from patient to patient2 might be interpreted misleadingly.... Please delete these 3 word.**

> Those words are now deleted.

\ **Liene 75-77 this is misleading. It should be mentioned that the observed cases occurred in solid organ transplant recipients but not in immunocompetent individuals.**

> We thank the Reviewer for this comment. It refers to Patient 1 only. Patients 2-4 were not transplanted patients and had a reduced immunocompetency due to their liver cirrhosis.

\ **Results section: The results section starts directly with the autopsy of the patient. This is unusual. Normally you first write "patient xx years old,sex.... transplanted xx years ago.. underlying disease... immunosuppression.....".**

> Detailed clinical information on Patient 1 is provided in the Supplementary File. This statement is now added at the beginning of the Results section.